# Distributed Learning with Strategic Users: A Repeated Game Approach

## Abstract

We consider a distributed learning setting where strategic users are incentivized, by a cost-sensitive fusion center, to train a learning model based on local data. The users are not obliged to provide their true gradient updates and the fusion center is not capable of validating the authenticity of reported updates. Thus motivated, we formulate the interactions between the fusion center and the users as repeated games, manifesting an under-explored interplay between machine learning and game theory. We then develop an incentive mechanism for the fusion center based on a joint gradient estimation and user action classification scheme, and study its impact on the convergence performance of distributed learning. Further, we devise an adaptive zero-determinant (ZD) strategy, thereby generalizing the celebrated ZD strategy to the repeated games with time-varying stochastic errors. Theoretical and empirical analysis show that the fusion center can incentivize the strategic users to cooperate and report informative gradient updates, thus ensuring the convergence.

## 1 Introduction

Distributed machine learning is becoming increasingly important in large-scale problems with data-intensive applications [18, 21, 25, 37]. Notably, federated learning has emerged as an attractive distributed computing paradigm that aims to learn an accurate model without collecting data from the owners and storing it in the cloud: The training data is kept locally on the computing devices which participate in the model training and report gradient updates (or its variants) based on local data [19].

In this work, we study a distributed learning scheme in which privacy-aware *users* train a global model with a *fusion center*. We consider the users to be rational, self-interested and risk-neutral. The users are not compelled to contribute their resources unconditionally, unless they are sufficiently rewarded, and the system may reach a noncooperative Nash equilibrium where the users do not participate in training. This departs from conventional distributed learning schemes where the agents directly follow the lead of the fusion center (FC)[1] and send their gradients. Since the users are strategic, a paramount objective for the FC is *to design an effective reward mechanism to incentivize self-interested users to provide informative gradient updates.* The repeated game enriches the distributed learning framework with the idea of many agents interacting within a common uncertain environment, and this framework provides a new perspective to specify how agents can strategically choose the learning updates how the resulting changes impact the performance of the learning efforts.

**Challenges and Contributions.** There are a number of challenges in distributed learning with strategic users. First, the users are not obliged to entirely dedicate their resources and they may not fulfill their roles in the training of the algorithm if it were not for their own interest. Secondly, the FC cannot directly validate data driven gradient updates due to their stochastic nature. The quality

---

[1]We refer to the fusion center as "she" and a user as "he".

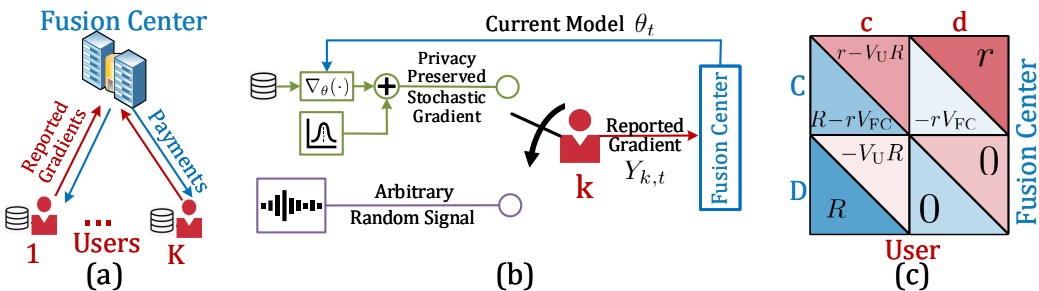

Figure 1: The fusion center (FC) trains the learning model with strategic users who are not obliged to report their gradients. (a) The objective of the FC is to incentivize users to cooperate by giving rewards so as to learn the model. (b) If the user is cooperative, he reports a privacy-preserved version of his gradient signal. Otherwise, the user is defective and sends an arbitrary uninformative signal. (c) The FC and the user each choose to cooperate or defect with respective payoffs as shown.

of the updates can vary over time and across the users since each user can control his own dataset. The interactions among users and the FC are repeated, and each user is capable of devising intricate strategies based on the past interactions. From a game-theoretic perspective, the fusion center's ability to reciprocate against non-cooperative user actions is significantly restricted since she cannot directly observe the user actions. Finally, the FC is not allowed to impose penalties on the users and positive rewards are the only options at her disposal to incentivize user participation. The work proposed here is, to the best of our knowledge, the first distributed learning framework to consider these challenges.

In this study, we model the interactions (in terms of gradient reporting and reward) between the FC and the users as repeated games, which intertwine with the updates in distributed learning. We propose a reward mechanism for the fusion center, based on an adaptive zero-determinant strategy, thereby generalizing the celebrated ZD strategy to the repeated games with time-varying stochastic errors. To tackle the challenge that the FC cannot directly verify the received reported gradients, we devise a gradient estimation and user action classification. Our findings demonstrate that, by employing adaptive ZD strategies, the FC can incentivize the strategic users to cooperate and report informative gradient updates, thus ensuing the convergence of distributed learning.

Detailed discussion on related work is relegated to Appendix A, due to space limitation.

## 2 Distributed Learning with Strategic Users as Repeated Games

We consider a distributed learning setting with $K$ strategic users $\mathcal{K} = \{1, \ldots, K\}$ and a fusion center (FC), and the optimization problem is given as follows:

$$\min_{\theta \in \mathbb{R}^n} F(\theta) := \frac{1}{K} \sum_{k=1}^{K} \mathbb{E}_{Z_k \sim \mathcal{D}} \big[ \mathcal{L}(\theta; Z_k) \big], \tag{1}$$

where $\mathcal{L}(\cdot)$ is the loss function. In each iteration, each user gets a mini-batch of $s$ i.i.d. samples from an unknown distribution $\mathcal{D}$, and computes the stochastic gradient signal as $X_{k,t} := \frac{1}{s} \sum_{i=1}^{s} \nabla_\theta \mathcal{L}(\theta_t; z_{k,t}^i)$, where $z_{k,t}^i$ is the $i^{\text{th}}$ sampled data of user $k$ at time $t$.

**Stage Game Formulation: Actions and Payoffs.** The action and the reported signal of user k in iteration $t$ are denoted with $B_{k,t} \in \{c, d\}$ and $Y_{k,t}$, respectively. As depicted in Fig. 1, a user is cooperative ($B_{k,t} = c$) if he is sending the privacy-preserved version of his gradient $X_{k,t}$. Otherwise, the user is defective and sends a noise signal $\Upsilon_{k,t} \sim \mathcal{N}(0, \Xi_t)$ independent of $X_{k,t}$:

$$Y_{k,t} = \begin{cases} X_{k,t} + N_{k,t}, & \text{if } B_{k,t} = c \text{ (cooperative);} \\ \Upsilon_{k,t}, & \text{if } B_{k,t} = d \text{ (defective).} \end{cases} \tag{2}$$

**Remark 1.** *Note that $N_{k,t}$ is independent of $X_{k,t}$ and $N_{k,t} \sim \mathcal{N}(\vec{0}, \nu_t^2 \mathbf{I})$. If $\|\nabla_\theta \mathcal{L}(\theta; z)\|_2 \leq \ell$ for all $\theta$ and $z$, then this privacy-preservation mechanism enjoys $\epsilon_t$-differential privacy, with $\epsilon_t = \ell^2 / s^2 \nu_t^2$ for mini-batch size $s$. The details are provided in Appendix.*

The payoff structure of a single interplay between the fusion center and a user is depicted in Fig 1b. In iteration $t$, when a user cooperates, he provides an information gain $R$ to the FC at his privacy cost $V_{\mathrm{U}}R$ with $0 < V_{\mathrm{U}} \leq 1$. When a user defects, he does not provide any information gain and does not incur any privacy cost. The FC may distribute rewards at the end of each iteration to incentivize the users. We denote the action of the FC toward user $k$ as $A_{k,t} \in \{C, D\}$. The FC is cooperative ($A_{k,t} = C$) if she makes a payment $r$ to the user at her cost $rV_{\mathrm{FC}}$ with $0 < V_{\mathrm{FC}} \leq 1$. The FC is defective ($A_{k,t} = D$), if she does not make any payment to the user. The factor $V_{\mathrm{FC}}$ captures the difference in the valuation of the reward between the FC and the user; for instance, the reward can be a coupon which may be redeemed in the future. Denote the FC's payoff vector by $\mathbf{S}_{\mathrm{FC}} = [R - rV_{\mathrm{FC}}, -rV_{\mathrm{FC}}, R, 0]$ and that of the users by $\mathbf{S}_{\mathrm{U}} = [r - V_{\mathrm{U}}R, r, -V_{\mathrm{U}}R, 0]$. In this paper, we only analyze the case where $R > rV_{\mathrm{FC}}$ and $r > V_{\mathrm{U}}R$. Otherwise, the FC or users do not have any incentive to cooperate.

The FC cannot observe the actions of the users and her realized payoffs. We assume that users do not communicate or collude with each other. They cannot observe the actions of other users and the actions of the FC toward other users. Next, we will discuss how to devise effective strategies for the FC to incentivize cooperative user action for the repeated game in a cost-effective manner.

**Repeated Games between Users and Fusion Center.** A salient feature of $2 \times 2$ repeated games is that players with longer memories of the history of the game have no advantage over those with shorter ones when each stage game is identically repeated infinite times [31]. Thus, without loss of generality, we assume the user strategies only depend on the outcomes of the last round. Let $q_1, q_2, q_3$ and $q_4$ denote the probabilities of cooperation for the user conditioned on the joint action pair of the previous iteration, that is $(A_{k,t-1}, B_{k,t-1})$, in the order of $(C, c), (C, d), (D, c)$ and $(D, d)$. The user's strategy vector is defined as $\mathbf{q} = [q_1, \ q_2, \ q_3, \ q_4]$.

Analogous to the user strategies, let $p_1, p_2, p_3$ and $p_4$ denote the probabilities of cooperation for the FC conditioned on $(A_{k,t-1}, B_{k,t})$, in the order of $(C, c), (C, d), (D, c)$ and $(D, d)$. The fusion center's strategy vector is defined as $\mathbf{p} = [p_1, \ p_2, \ p_3, \ p_4]$. The joint action pair of the user and the FC is considered as the state of the game in iteration $t$: $(A_{k,t}, B_{k,t})$. The strategy vectors $\mathbf{p}$ and $\mathbf{q}$ imply a Markov state transition matrix as follows:

$$
\Omega = \begin{bmatrix}
q_1 p_1 & (1 - q_1) p_2 & q_1 (1 - p_1) & (1 - q_1)(1 - p_2) \\
q_2 p_1 & (1 - q_2) p_2 & q_2 (1 - p_1) & (1 - q_2)(1 - p_2) \\
q_3 p_3 & (1 - q_3) p_4 & q_3 (1 - p_3) & (1 - q_3)(1 - p_4) \\
q_4 p_3 & (1 - q_4) p_4 & q_4 (1 - p_3) & (1 - q_4)(1 - p_4)
\end{bmatrix}.
\tag{3}
$$

Let $\Lambda^*$ be the stationary vector of the transition matrix $\overline{\Omega}$, i.e., $\Lambda^* = \Lambda^* \overline{\Omega}$. We can find the expected payoffs of the FC and the user in the stationary state as $s_{\mathrm{FC}}^* = \Lambda^* \mathbf{S}_{\mathrm{FC}}^\top$ and $s_{\mathrm{U}}^* = \Lambda^* \mathbf{S}_{\mathrm{U}}^\top$. The FC sets her strategy $\mathbf{p}$ satisfying, for some real values $\varphi_0$, $\varphi_1$ and $\varphi_2$, the equation

$$
[p_1 - 1, \ p_2 - 1, \ p_3, \ p_4] = \varphi_0 \mathbf{S}_{\mathrm{FC}} + \varphi_1 \mathbf{S}_{\mathrm{U}} + \varphi_2 \mathbf{1}.
\tag{4}
$$

This class of strategies are called zero-determinant (ZD) strategies, which enforce a linear relation between the expected payoffs, given by $\varphi_0 s_{\mathrm{FC}}^* + \varphi_1 s_{\mathrm{U}}^* + \varphi_2 = 0$, regardless of the user strategy [31].

**Remark 2.** *The ZD strategy is a powerful tool to incentivize the users cooperation for the FC because she can unilaterally set $s_{\mathrm{U}}^*$ or establish an extortionate linear relation between $s_{\mathrm{U}}^*$ and $s_{\mathrm{FC}}^*$. Against such an FC strategy, the user's best response which maximizes his payoff is full cooperation, $\mathbf{q}^* = [1\,1\,1\,1]$. The details are provided in Appendix C.*

Against the FC who is equipped with the ZD strategy, the user can increase his expected payoff only by cooperating more often, and consequently his best response is full cooperation. Assuming that there are sufficiently many participating users, the FC has the absolute leverage against any single user who tries to negotiate with her. Nevertheless, the FC cannot directly employ the ZD strategy since she cannot observe the true actions of the users. In the next section, we will study the use of ZD strategy can be extended in the scope of distributed learning.

# 3 Distributed Stochastic Gradient Descent with Strategic Users

For the ease of exposition, in this paper we focus on an interesting variant of the classical stochastic gradient descent algorithm using the gradient signals reported by strategic users (SGD-SU). In each iteration, the FC collects the reported gradients of the users and update the model as follows:

$$
\theta_t = \theta_{t-1} - \eta_t \cdot \widehat{m}_t(\mathbf{Y}_t),
\tag{5}
$$

**Algorithm 1:** Stochastic Gradient Descent with Strategic Users (SGD-SU)

---

**1 for** $t = 1, 2, \ldots, T - 1$ **do**

**2**  *Fusion Center:* broadcast the current iterate $\theta_{t-1}$ to all the users

**3**  **forall** $k \in \{1, 2, \ldots, K\}$ **do**

**4**   *User k:* compute the gradient $X_{k,t}$ and $Y_{k,t} \leftarrow \begin{cases} X_{k,t} + N_{k,t} & \text{cooperative action,} \\ \Upsilon_{k,t} & \text{defective action,} \end{cases}$

**5**  *Fusion Center:* form the gradient estimate $\widehat{m}_t(\mathbf{Y}_t) \leftarrow \frac{1}{K(\Lambda_1 \Omega^{t-1})\mathbf{q}^\top} \sum_{k=1}^{K} Y_{k,t}$

**6**  update model parameter $\theta_t \leftarrow \theta_{t-1} - \eta_t \widehat{m}_t(\mathbf{Y}_t)$

**7**  classify the users $\widehat{B}_{k,t}(\widehat{m}_t, Y_{k,t}) \leftarrow \begin{cases} \hat{c} & \text{(cooperative)} \quad \text{if} \quad Y_{k,t}^\top \widehat{m}_t > \|\frac{1}{2}\widehat{m}_t\|_2^2 \\ \hat{d} & \text{(defective)} \qquad \text{else} \end{cases}$ (7)

**8**  compute the detection and false alarm probabilities using (8) and (11)

**9**  compute the adaptive strategies (9) and distribute the rewards accordingly

---

where $\mathbf{Y_t} = [Y_{1,t} \ldots Y_{K,t}]$, $\eta_t$ is the step size and $\widehat{m}_t$ is the gradient estimator. The FC cannot directly observe user actions and verify the reported gradients. This gives rise to two coupled challenges:

• The gradient estimator $\widehat{m}_t$ should be resilient against the uninformative reports of defective users.

• Although the ZD strategies are powerful tools to incentivize user cooperation, the FC cannot directly employ a ZD strategy because she cannot observe the users' actions.

To tackle these difficulties, we will first introduce a gradient estimation and user classification scheme and discuss the impact of user action classification errors on the dynamics of repeated games. As outlined in Algorithm 1. we will develop adaptive FC strategies which generalize the classical ZD strategies to the repeated games with time-varying stochastic errors.

### 3.1 Joint Gradient Estimation and User Action Classification

The stochastic gradients can be decomposed as $X_{k,t} = m_t + W_{k,t}$ where $m_t := \nabla_\theta F(\theta_t)$ is the population gradient and $W_{k,t}$ is the zero-mean noise term [30]. The unknown parameter $m_t$ is the mean of the reported gradient $Y_{k,t}$ when the user is cooperative ($B_{k,t} = c$). The defective users send zero-mean random noise as their reported gradients. The FC needs to classify the reported gradients and obtain an estimate of $m_t$ for the SGD-SU update in (5). These two problems are coupled with each other, and the joint scheme is, therefore, comprised of a gradient estimator $\widehat{m}_t$, and a classification rule $\widehat{B}_{k,t}$. To tackle this difficult problem, we first investigate gradient estimation.

Let $\Lambda_1$ be the initial state distribution of the games between the users and the FC. A modified empirical mean based gradient estimator can be employed as follows:

$$\widehat{m}_t(\mathbf{Y}_t) := \frac{1}{K(\Lambda_1 \Omega^{t-1})\mathbf{q}^\top} \sum_{k=1}^{K} Y_{k,t}. \tag{6}$$

It is easy to verify that $\widehat{m}_t(\cdot)$ is an unbiased estimator if the FC is able to employ her strategies $\mathbf{p}$ without any errors and the state distribution of the repeated games are governed by the state transition matrix $\Omega$ as in (3) without any perturbations.

Using the gradient estimator $\widehat{m}_t(\cdot)$, the FC can form the user action classification rule as

$$\widehat{B}_{k,t}(\widehat{m}_t(\mathbf{Y}_t), Y_{k,t}) = \begin{cases} \widehat{c} & \text{if} \quad Y_{k,t}^\top \widehat{m}_t > \frac{1}{2}\|\widehat{m}_t\|^2, \\ \widehat{d} & \text{else;} \end{cases} \tag{7}$$

where $\hat{d}$ (or $\hat{c}$) is the defective (or cooperative) label. The noise in the stochastic gradients, $W_{k,t}$, can be approximated as a zero mean Gaussian r.v. [17, 22, 26, 36]. Recall from (2) that cooperative users send the privacy-preserved versions of their gradient. This implies $Y_{k,t} \sim \mathcal{N}(m_t, \Sigma_t)$, given $B_{k,t} = c$, where $\Sigma_t := \text{cov}[W_{k,t}] + \nu_t^2 \text{I}$. Thus, the detection and false alarm probabilities of the classifier, denoted by $\Phi_t$ and $\Psi_t$ respectively, can be found as

$$\Phi_t = 1 - \mathcal{Q}\left(\frac{m_t^\top \widehat{m}_t - \frac{1}{2}\|\widehat{m}_t\|^2}{\sqrt{\widehat{m}_t^\top \Sigma_t \widehat{m}_t}}\right) \quad \text{and} \quad \Psi_t = \mathcal{Q}\left(\frac{\frac{1}{2}\|\widehat{m}_t\|^2}{\sqrt{\widehat{m}_t^\top \Xi_t \widehat{m}_t}}\right). \tag{8}$$

**Remark 3.** *The linear classifier* (7) *is an effective tool under the homoscedasticity assumption. If that is violated, the FC can employ different classifiers. The details are provided in Appendix for the Classifier Design.*

In the next subsection, we discuss how the FC can devise her strategies building on the joint gradient estimation and user action classification scheme.

## 3.2 Adaptive Strategies for Fusion Center

Although the ZD strategies, $\mathbf{p}$, provide the FC an efficient and powerful mechanism to encourage the user's cooperation; the FC cannot directly use $\mathbf{p}$ since they are conditioned on the user's action, $B_{k,t}$, which is not observable to her. Alternatively, the FC can use the classification results after carefully *adapting* her strategies to mitigate the adverse effects of inevitable classification errors. Let $\pi_{t,1}, \pi_{t,2}, \pi_{t,3}$ and $\pi_{t,4}$ denote the probabilities of cooperation for the FC conditioned on $(A_{k,t-1}, \widehat{B}_{k,t})$, in the order of $(C, \hat{c}), (C, \hat{d}), (D, \hat{c})$ and $(D, \hat{d})$. These are referred to as *adaptive* strategies and the FC sets these probabilities satisfying the following system of equations:

$$p_1 = \pi_{t,1}\Phi_t + \pi_{t,2}(1 - \Phi_t), \quad p_2 = \pi_{t,1}\Psi_t + \pi_{t,2}(1 - \Psi_t),$$
$$p_3 = \pi_{t,3}\Phi_t + \pi_{t,4}(1 - \Phi_t), \quad p_4 = \pi_{t,3}\Psi_t + \pi_{t,4}(1 - \Psi_t).$$

Suppose $\frac{\Phi_t}{\Psi_t} \geq \frac{p_1}{p_2}$ and $\frac{\Phi_t}{\Psi_t} \geq \frac{p_3}{p_4}$. Then the unique solution to the system above is given by

$$\pi_{t,1} = \frac{p_1(1 - \Psi_t) - p_2(1 - \Phi_t)}{\Phi_t - \Psi_t}, \quad \pi_{t,2} = \frac{p_2\Phi_t - p_1\Psi_t}{\Phi_t - \Psi_t}, \tag{9a}$$

$$\pi_{t,3} = \frac{p_3(1 - \Psi_t) - p_4(1 - \Phi_t)}{\Phi_t - \Psi_t}, \quad \pi_{t,4} = \frac{p_4\Phi_t - p_3\Psi_t}{\Phi_t - \Psi_t}. \tag{9b}$$

**Remark 4.** *If the FC directly employed the ZD strategies without any adaptation, i.e., she cooperates with probability $p_i$ conditioned on classification output; the repeated games may not converge to the stationary state $\Lambda^*$ and a linear relation between the expected payoffs* (4) *may not be enforced because the classification errors yield an additive disturbance on the state transition matrix as follows*

$$\Omega - (p_1 - p_2)\left\{\mathbf{q}^\top[1 - \Phi_t\ 0\ 1 - \Phi_t\ 0] + (\mathbf{1} - \mathbf{q})^\top[0\ \Psi_t\ 0\ \Psi_t]\right\}. \tag{10}$$

*Adaptive strategies* (9) *cancel out this adverse disturbance on the dynamics of the repeated games.*

In the absence of classification errors ($\Phi_t = 1$ and $\Psi_t = 0$), the adaptive strategies reduce to the ZD strategies, i.e., $\boldsymbol{\pi}_t = \mathbf{p}$. Classification errors force the FC to be more *retaliatory* than dictated by the ZD strategy $\mathbf{p}$, i.e., $\pi_{t,1} > p_1$, $\pi_{t,3} > p_3$, $\pi_{t,2} < p_2$ and $\pi_{t,4} < p_4$. In general, detection and false alarm probabilities, $\Phi_t$ and $\Psi_t$, are time-varying; thus the adaptive strategies also change over time.

## 3.3 The Impact of Estimation Errors on Repeated Game Dynamics

The proposed adaptive strategies (9) requires the knowledge of detection probability, $\Phi_t$. However, the FC cannot exactly compute $\Phi_t$ using (8) since she does not have the knowledge of $m_t$. Instead, she can form her estimate $\widehat{\Phi}_t$ using $\widehat{m}_t$:

$$\widehat{\Phi}_t = 1 - \mathcal{Q}\left(\frac{\frac{1}{2}\|\widehat{m}_t\|^2}{\sqrt{\widehat{m}_t^\top \Sigma_t \widehat{m}_t}}\right) \tag{11}$$

Due to the inevitable gradient estimation errors, in general, we have $\widehat{\Phi}_t \neq \Phi_t$. As a result, the FC cannot exactly employ the adaptive FC strategies dictated by Eq. 9. With several steps of variable substitutions, this yields an additive perturbation on the state transition matrix as follows:

$$\widetilde{\Omega}_t = \Omega + V_t\Omega^\perp \text{ with } V_t := \frac{\widehat{\Phi}_t - \Phi_t}{\widehat{\Phi}_t - \Psi_t} \text{ and } \Omega^\perp := (p_1 - p_2)\mathbf{q}^\top[-1\ 0\ 1\ 0]. \tag{12}$$

Let $\tilde{\Lambda}_t$ be the probability distribution over the state space of the games $\{Cc, Cd, Dc, Dd\}$ at the start of iteration $t$. According to (12), the state distributions follow the transition rule such that

$$\widetilde{\Lambda}_{t+1} = \widetilde{\Lambda}_t\widetilde{\Omega}_t = \widetilde{\Lambda}_t\left(\Omega + V_t\Omega^\perp\right).$$

Note that $\Lambda_t$ can be considered as the state distribution of the repeated games in the absence of perturbations on the state transition matrix. For the FC, $\Lambda_t$ is the designed state distribution in which the ZD strategy dominates against any user strategy.

Next, we study the time-varying perturbation terms. Using (8) and (11), $V_t$ can be found as[2]:

$$V_t = \frac{\widehat{\Phi}_t - \Phi_t}{\widehat{\Phi}_t - \Psi_t} = \frac{\mathcal{Q}\left(\frac{\widehat{m}_t^\top\left(m_t - \frac{1}{2}\widehat{m}_t\right)}{\sqrt{\widehat{m}_t^\top \Sigma_t \widehat{m}_t}}\right) - \mathcal{Q}\left(\frac{\frac{1}{2}\|\widehat{m}_t\|^2}{\sqrt{\widehat{m}_t^\top \Sigma_t \widehat{m}_t}}\right)}{1 - \mathcal{Q}\left(\frac{\frac{1}{2}\|\widehat{m}_t\|^2}{\sqrt{\widehat{m}_t^\top \Sigma_t \widehat{m}_t}}\right) - \mathcal{Q}\left(\frac{\frac{1}{2}\|\widehat{m}_t\|^2}{\sqrt{\widehat{m}_t^\top \Xi_t \widehat{m}_t}}\right)} = \frac{\mathcal{Q}\left(\frac{\frac{\widehat{m}_t(m_t - \widehat{m}_t)}{\|m_t\|} + \frac{1}{2}\|\widehat{m}_t\|}{\sqrt{\text{Ray}(\Sigma_t, \widehat{m}_t)}}\right) - \mathcal{Q}\left(\frac{\frac{1}{2}\|\widehat{m}_t\|}{\sqrt{\text{Ray}(\Sigma_t, \widehat{m}_t)}}\right)}{1 - \mathcal{Q}\left(\frac{\frac{1}{2}\|\widehat{m}_t\|}{\sqrt{\text{Ray}(\Sigma_t, \widehat{m}_t)}}\right) - \mathcal{Q}\left(\frac{\|m_t\|}{\sqrt{\text{Ray}(\Xi_t, \widehat{m}_t)}}\right)}.$$

In the presence of these perturbations, to establish stability guarantees on the dynamics of the repeated games, we impose the following assumption on the norm of the gradient estimator:

**Assumption 1.** *We assume that* $\|\widehat{m}_t\| \geq \max\left\{2\sqrt{\text{Ray}(\widehat{m}_t, \Sigma_t)}, 2\sqrt{\text{Ray}(\widehat{m}_t, \Xi_t)}, \sqrt{\widehat{m}_t^\top(m_t - \widehat{m}_t)}\right\}.$

Note that these conditions are primarily associated to the accuracy of the linear classifier (7) which operates effectively when the mean vectors of the classes are sufficiently separated. The following result indicates that, due to the perturbations on the state transition matrix, the real state distribution $\tilde{\Lambda}_t$ is a noisy version of $\Lambda_t$.

**Lemma 1.** *Let $\Lambda_1$ denote the initial state distributions of the games between the FC and the users. Under Assumption 1, we have that*

$$\tilde{\Lambda}_t = \Lambda_t + \Lambda_1 \sum_{i=1}^{t-1} V_i \Omega^{i-1} \Omega^\perp \Omega^{t-1-i}. \tag{13}$$

This noise on the state distributions will manifest as a novel bias term in the gradient estimation. In the next subsection, we will provide the convergence analysis of SGD-SU which will mainly focus on the characterization of this bias term.

## 3.4 Convergence Results

In this section, we provide the convergence guarantee for SGD-SU (5). Let $\mathcal{F}_t$ denote the $\sigma$-algebra, generated by $\{\theta_1, \mathbf{Y}_i, i < t\}$. In particular, $\mathcal{F}_t$ should be interpreted as the history of SGD-SU up to iteration $t$, just before $\mathbf{Y}_t$ is generated. Thus, conditioning on $\mathcal{F}_t$ can be thought of as conditioning on $\{\theta_1, \tilde{\Lambda}_1, \mathbf{Y}_1, \ldots, \theta_{t-1}, \tilde{\Lambda}_{t-1}, \mathbf{Y}_{t-1}, \theta_t, \tilde{\Lambda}_t\}$. For convenience, denote $\mathbb{E}_t[\cdot] := \mathbb{E}_t[\cdot|\mathcal{F}_t]$. Observe that, we can decompose the gradient estimator $\widehat{m}_t$ as follows:

$$\widehat{m}_t(\cdot) = m_t(1 + \zeta_t) + \mathcal{E}_t, \tag{14}$$

where $\zeta_t$ is the estimation bias term due to the perturbations on the state transition matrix, given by

$$\zeta_t = \frac{1}{m_t}\left(\mathbb{E}_t[\widehat{m}_t] - m_t\right) = \frac{\sum_{k=1}^K \text{P}(B_{k,t} = c|\mathcal{F}_t)}{K(\Lambda_t \mathbf{q}^\top)} - 1$$

and $\mathcal{E}_t$ is the estimation noise term, given by $\mathcal{E}_t = \widehat{m}_t - \mathbb{E}_t[\widehat{m}_t]$. Conditioned on $\mathcal{F}_t$, the probability of a user taking the cooperative action, in iteration $t$, is given by $\text{P}(B_{k,t} = c|\mathcal{F}_t) = \tilde{\Lambda}_t \mathbf{q}^\top$. The bias term, $\zeta_t$, can be found as follows:

$$\zeta_t = \frac{\tilde{\Lambda}_t \mathbf{q}^\top}{\Lambda_t \mathbf{q}^\top} - 1. \tag{15}$$

From Lemma 1 and (15), it is clear that the perturbations on the state transition matrix (12), directly translates into a bias in the gradient estimation rule.

To establish convergence guarantees for the SGD-SU in (5), $\Lambda_t \mathbf{q}^\top$ and $\tilde{\Lambda}_t \mathbf{q}^\top$ must meet the following criteria during the course of the algorithm:

**Assumption 2.** *We assume that $\Lambda_t \mathbf{q}^\top > \frac{1}{2}$ and $\tilde{\Lambda}_t \mathbf{q}^\top > 0$, for all $t \in \{1, 2, \ldots, T\}$.*

---

[2]The Rayleigh's quotient for a symmetric matrix $M$ and nonzero vector $x$ is defined as $\text{Ray}(M, x) = \frac{x^\top M x}{x^\top x}$

The first condition $\Lambda_t \mathbf{q}^\top \geq 0.5$ is very mild in the sense that it merely requires that the probability of user cooperation dictated by the memory-1 strategies $\mathbf{p}$ and $\mathbf{q}$ ($1 \times 4$ vectors), in the absence of perturbations, is larger than 0.5. The second condition $\widetilde{\Lambda}_t \mathbf{q}^\top > 0$ states that, in the presence of perturbations, the probability of user cooperation is always positive[3].

By Assumption 2, there exists a positive constant $H_T$ such that

$$0 < |\zeta_t| < H_T < 1, \ \forall t \in \{1, \ldots, T\}. \tag{16}$$

Further, we have the following lemma characterizing the properties of estimation noise.

**Lemma 2.** *Conditioned on $\mathcal{F}_t$, the estimation noise in iteration $t$, denoted $\mathcal{E}_t$, is a zero-mean random vector with the mean square error given by*

$$\mathbb{E}_t[\|\mathcal{E}_t\|^2] = \frac{1}{K\left(\Lambda_t \mathbf{q}^\top\right)} \left( (\zeta_t + 1)\mathrm{tr}\left(\Sigma_t - \Xi_t\right) + \frac{1}{\Lambda_t \mathbf{q}^\top}\mathrm{tr}\left(\Xi_t\right) \right). \tag{17}$$

By (16) and (17), we have that

$$\mathbb{E}_t\left[\|\mathcal{E}_t\|^2\right] \leq \frac{E_T}{K} \text{ with } E_T := \frac{1}{\Lambda_t \mathbf{q}^\top}\left[ \left(H_T + 1\right)\mathrm{tr}(\Sigma_t - \Xi_t) + \frac{1}{\Lambda_t \mathbf{q}^\top}\mathrm{tr}(\Xi_t) \right]. \tag{18}$$

We impose the following assumption on the objective function, which is standard for performance analysis of stochastic gradient-based methods [3, 28].

**Assumption 3.** *The objective function $F$ and the SGD-SU satisfy the following:*

*(i) $F$ is $L-$smooth, that is, $F$ is differentiable and its gradient is $L-$Lipschitz:*
$$\|\nabla F(\theta) - \nabla F(\theta')\| \leq L\|\theta - \theta'\|, \ \forall \theta, \theta' \in \mathbb{R}^n.$$

*(ii) The sequence of iterates $\{\theta_t\}$ is contained in an open set over which $F$ is bounded below by a scalar $F_{\inf}$.*

Our next result describes the behavior of the sequence of gradients of $F$ when fixed step sizes are employed.

**Theorem 1.** *Under Assumptions 2 and 3, suppose that the SGD-SU (5) is run for $T$ iterations with a fixed stepsize $\bar{\beta}$ satisfying*

$$0 < \bar{\beta} \leq \frac{1}{L(1 + H_T)}. \tag{19}$$

*Then, the SGD algorithm with strategic users satisfies that*

$$\mathbb{E}\left[ \frac{1}{T}\sum_{t=1}^{T}\|\nabla F(\theta_t)\|^2 \right] \leq \frac{LE_T}{K(1 - H_T)} + \frac{2(F(\theta_1) - F_{\inf})}{\bar{\beta}T(1 - H_T)}.$$

Theorem 1 illustrates the impact of the perturbations on the state transition matrix (12) on the convergence rate of SGD-SU. When $H_T$ is close to 0, SGD-SU performs similar to the basic minibatch SGD. On the other hand, if $H_T$ is close to 1, the optimality gap may be large. Our next result will characterize the gradient estimation bias term $\zeta_t$. First, we have the following assumption on the state transition matrix $\Omega$.

**Assumption 4.** *The state transition matrix $\Omega$ can be diagonalized as $\Omega = \Gamma \mathcal{U} \Gamma^{-1}$ with $\mathcal{U}$ has the eigenvalues of $\Omega$ in descending order of magnitude: $1 \geq |u_2| \geq |u_3| \geq |u_4| \geq 0$.*

Denote the element of $\Gamma^{-1}$ in the $i^{\text{th}}$ row and $j^{\text{th}}$ column as $\Gamma_{ij}^{-1}$. Denote the four rows of $\Gamma^{-1}$ by $\vec{\gamma}_1, \ldots, \vec{\gamma}_4$. Next, we define $\delta$ as

$$\delta := \left( \max_{j \in \{2,3,4\}} \left|\Gamma_{3j} - \Gamma_{1j}\right| \right) \left( \max_{j \in \{2,3,4\}} \left|\vec{\gamma}_j \mathbf{q}^\top\right|^2 \right).$$

Further, the first order Taylor approximation of the scalar variable $V_t$ can be found as follows:

$$V_t = \frac{m_t^\top (\widehat{m}_t - m_t)}{\|m_t\|^2} h_t(m_t) \text{ with } h_t(m_t) := \frac{\frac{\|m_t\|}{\sqrt{2\pi \mathrm{Ray}(\Sigma_t, m_t)}}\exp\left( -\frac{1}{8}\frac{\|m_t\|^2}{\mathrm{Ray}(\Sigma_t, m_t)} \right)}{1 - \mathcal{Q}\left( \frac{\|m_t\|}{2\sqrt{\mathrm{Ray}(\Sigma_t, m_t)}} \right) - \mathcal{Q}\left( \frac{\|m_t\|}{2\sqrt{\mathrm{Ray}(\Xi_t, m_t)}} \right)}. \tag{20}$$

---

[3]A sufficient condition for this requirement is that user strategies are *forgiving* in nature, i.e., $q_1, q_2, q_3, q_4 > 0$.

Define $h_t^{\max} := \max_{i \in \{1,\ldots,t\}} h_i(m_i)$. Our next result indicates that, the estimation bias term $\zeta_t$ can be found in terms of the past gradient estimation errors.

**Theorem 2.** *Under Assumptions 1, 2 and 4, the gradient estimation bias term $\zeta_t$, can be found as*

$$\zeta_t = (p_1 - p_2) \sum_{i=1}^{t-1} \frac{\Lambda_i \mathbf{q}^\top}{\Lambda_t \mathbf{q}^\top} \frac{m_i^\top \mathcal{E}_i}{\|m_i\|^2} h_i(m_i) \Delta_{i,t} \tag{21a}$$

*with*

$$|\Delta_{i,t}| \le \delta |u_2|^{t-1-i} + \delta^2 h_{t-1}^{\max} |u_2|^{t-2-i}(t-i-1). \tag{21b}$$

*Further, for some $0 < \eta < 1$ we have*

$$\mathrm{P}\left(|\zeta_t| < \eta | \alpha_1, \ldots, \alpha_{t-1}\right) > 1 - \frac{\sum_{i=1}^{t-1} \alpha_i^2}{K \eta^2} \tag{22a}$$

*with*

$$\alpha_i^2 = \frac{2\left|(\nu_i^2 - \xi_i^2) + \frac{m_i^\top \Sigma_i m_i}{\|m_i\|^2}\right| + \frac{\xi_i^2}{\Lambda_i \mathbf{q}^\top}}{\|m_i\|^2 \left(\Lambda_i \mathbf{q}^\top\right)} \left[\frac{\Lambda_i \mathbf{q}^\top}{\Lambda_t \mathbf{q}^\top}\right]^2 h_i^2 \Delta_{i,t}^2. \tag{22b}$$

Note that Eq. (21) indicates that, the estimation bias term $\zeta_t$ can be expanded in terms of past gradient estimation errors. We prove that the absolute values of the coefficients, $|\Delta_{i,t}|$'s, are bounded as

$$|\Delta_{i,t}| \le \delta |u_2|^{t-1-i} + \delta^2 h_{t-1}^{\max} |u_2|^{t-2-i}(t-i-1),$$

where $u_2$ is the eigenvalue of $\Omega$ with the second highest absolute value. Since $\Omega$ is a row stochastic matrix, $|u_2| \le 1$. When $|u_2|$ is strictly less than 1, $\Delta_{i,t}$'s decay fast as $t-i$ grows. This can also be interpreted as the impact of past gradient estimation errors fade away quickly. Using this result, in Eq.(22), we derive a high probability upper bound on the estimation bias term $\zeta_t$.

# 4   Experiments

In this section, we evaluate the performance of SGD-SU (5) using real-life datasets. All the results in the preceding section assert convergence for the SG method (5) under the assumption that the FC can access $\Sigma_t$ and $\Xi_t$. In a real-life machine learning setting with strategic users, this information may not be available to the FC. For convenience, define $\widehat{\mathcal{K}}_t^c$ and $\widehat{\mathcal{K}}_t^d$ as the sets of users who are classified as cooperative $(\hat{c})$ and defective $(\hat{d})$ at iteration $t$. Based on the user action classification, the FC can form her estimates for the covariance matrices under the cooperative and defective actions as follows:

$$\widehat{\Sigma}_t = \frac{1}{|\widehat{\mathcal{K}}_t^c|} \sum_{k \in \widehat{\mathcal{K}}_t^c} \left(Y_{k,t} - \bar{Y}_t^c\right)\left(Y_{k,t} - \bar{Y}_t^c\right)^\top \text{ and } \quad \widehat{\Xi}_t = \frac{1}{|\widehat{\mathcal{K}}_t^d|} \sum_{k \in \widehat{\mathcal{K}}_t^d} \left(Y_{k,t} - \bar{Y}_t^d\right)\left(Y_{k,t} - \bar{Y}_t^d\right)^\top, \tag{23}$$

where $\bar{Y}_t^c = \frac{1}{|\hat{\mathcal{K}}_t^c|} \sum_{k \in \hat{\mathcal{K}}_t^c} Y_{k,t}$ and $\bar{Y}_t^d = \frac{1}{|\hat{\mathcal{K}}_t^d|} \sum_{k \in \hat{\mathcal{K}}_t^c} Y_{k,t}$.

In our first set of experiments, we consider a binary logistic classification problem and use the KDD-Cup 04 dataset [6]. The goal of binary logistic classification experiments is to learn a classification rule that differentiates between two types of particles generated in high energy collider experiments based on 78 attributes [6]. In our second set of experiments, we consider a neural network trained on the MNIST dataset. The number of users is chosen as $K = 50$ and mini-batch size is $s = 10$. In the experiments, we have tested the performance of two different ZD strategies, namely *equalizer* and *extortion*[31].

For the logistic classification problem, Fig. 4a and 4b, depict the optimality gap under four different user strategies: $\mathbf{q} = [0.9\ 0.15\ 0.9\ 0.15]$ (stubborn), $\mathbf{q} = [0.9\ 0.9\ 0.15\ 0.15]$ (tit-for-tat, ), $\mathbf{q} = [0.9\ 0.15\ 0.15\ 0.9]$ (win-stay-lose-switch) and $\mathbf{q} = [0.9\ 0.9\ 0.9\ 0.9]$ (full cooperation). For the full cooperation, coin toss, tit-for-tat and stubborn user strategies, SGSU converges quickly. For Pavlov user strategies, SGSU can eventually approach, albeit more slowly than other cases. Fig 4c and 4d illustrate the probability of user cooperation, $\widetilde{\Lambda}_t \mathbf{q}^\top$, across different user strategies. The experimental results validate Lemma 1 and the empirical user cooperation probabilities match the theoretical except when the users are Pavlov. Unsurprisingly, when the users follow full cooperation (or coin toss) strategy, they cooperate with probability 0.9 (or 0.5) regardless of the actual states of the repeated

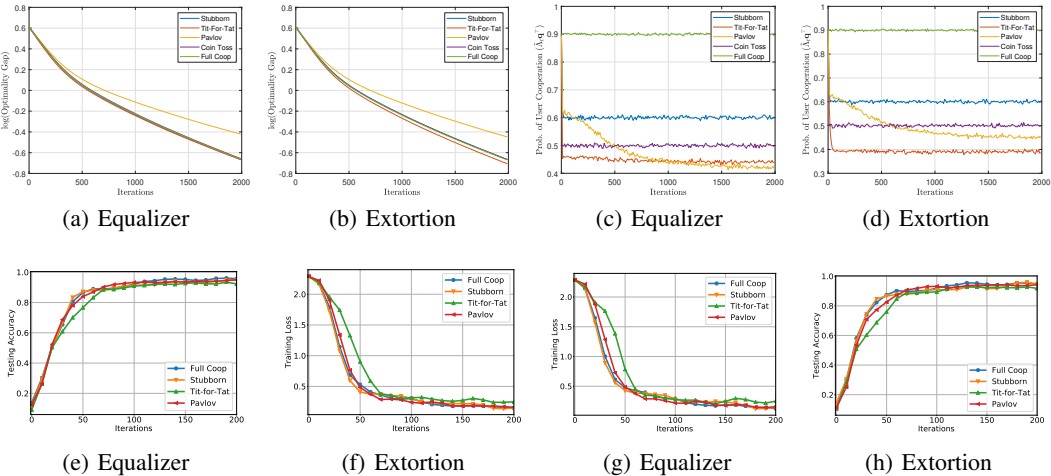

Figure 2: Stochastic Descent Algorithm with Strategic Users

games. For the cases with stubborn and tit-for-tat users, the games quickly converge to the steady state distribution. Interestingly, for the cases with Pavlov users, the probability of user cooperation decreases over time. This is associated to the performance of the linear classifier. For the image classification problem, Fig 4e-h depict the training loss and testing accuracy across iterations for different FC and user strategies. In all experiments, SGSU converges in the presence of strategic users. Further details regarding the Experimental results are relegated to Appendix.

## 5 Future Directions

In this work, we study a distributed learning framework where strategic users train a learning model with a fusion center. The main objective of the FC is to encourage users to be cooperative by distributing rewards. Based on this, we devise a reward mechanism for the FC based on the ZD-strategies. Further, we examine the performance of SGD algorithm in the presence of strategic users. Our findings reveal that the algorithm has provable convergence and our empirical results verify our theoretical analysis.

We are also working on the development of robust estimation tools in distributed learning with strategic users. The geometric median is a reliable estimation technique when the collected data contain outliers of large magnitude [10, 14, 24, 27]:

$$\mathrm{Med}(\mathbf{Y}_t) := \arg \min_{y \in \mathbb{R}^n} \sum\nolimits_{k=1}^{K} \|y - Y_{k,t}\|_2. \tag{24}$$

The FC can use $\mathrm{Med}$ as a robust gradient estimator, especially when the variance of the uninformative signals, $\xi_t^2$, reported by the defective users, is very high. The geometric median (24) can be computed by the Weiszfeld's algorithm [34, 35], which is a special case of iteratively reweighted least squares. In contrast, with the knowledge of $\mathbf{q}$, the modified sample mean estimator (6) allows the FC to trade robustness for overall tractability of the algorithm with reduced computational complexity.

The linear classifier is vulnerable to vanishing gradients as the stochastic gradient descent algorithm with strategic users (SGD-SU) converges to $\theta^*$. This can be addressed by modifying the classifier to incorporate the information contained in the norm of the reported gradients. Furthermore, we discuss how to extend the convergence guarantee for SGSU to allow heterogeneous user strategies. The details are presented in Appendix.

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
