# OpenReview forum: "Distributed Learning with Strategic Users: A Repeated Game Approach"
_NeurIPS.cc/2021/Conference — NeurIPS 2021 Submitted_

### Official Review · Reviewer_mQAc · 2021-07-06

**Rating:** 5
**Confidence:** 3

**Summary:**

This paper studies the incentives in distributed learning, where the users can choose to join the cooperation or not. The paper formalizes this problem as a two-player game between the fusion center and the user, designs a strategy to learn and provides a convergence guarantee. The contributions include: 1. formalizing of a reward mechanism between FC and users; 2. proposing a strategy for the FC to make users join the cooperation, where FC cannot observe users' actions; 3. providing theoretical guarantee on the convergence rate.

**Limitations And Societal Impact:**

No. The authors are recommended to add a discussion on the societal impact.

**Main Review:**

Originality: The task is new. The problem formalization and the mechanism proposed are novel.

Quality: The submission is technically sound. But the assumptions are somewhat unreasonable to me and I would like to know if these assumptions are necessary and non-vacuous.

Specifically, for Assumption 1, it might be uncommon to make assumptions over the gradient estimate, which is a r.v. dependent on the objective function and the noise distributions. I am curious if the assumption could be made directly over the objective function and the distributions instead of the gradient estimate. For Assumption 2, $\tilde{\Lambda}_t$ depends on $\Lambda_t$, the objective function and the noise distributions. Similar to Assumption 1, it looks quite strange to make direct assumptions over $\tilde{\Lambda}_t$.  Also, before Assumption 2, the authors state that ``" $\Lambda_t q^\top$ and $\tilde{\Lambda}_t q^\top$ must meet the following criteria ...". Is Assumption 2 a necessary condition?

Besides, the gradient estimator $\hat m_t$ depends on $q$. It is unclear to me what $q$ is.  Is it the user's initial strategy? Is it deterministic?

Clarity: Some notations are unclear. It would be better if the paper has more explanation on the assumptions.

Significance: The problem studied by this paper is interesting and important.

Other comments:
1. What is $Q$ in Eq. (8)?
2. Is $\hat {m}_t^\top (m_t - \hat m_t)$ in Assumption 1 always positive? If not, $\sqrt{\hat m_t^\top (m_t - \hat m_t)}$  is not well-defined.
3. Typo: $Ray(\hat m_t, \Sigma_t)$ should be $Ray(\Sigma_t,\hat m_t)$.

**Time Spent Reviewing:**

7

---

> ### Author Response · Authors · 2021-08-14
> **Author Response to Reviewer mQAc**
>
> The authors would like to express sincere thanks to the reviewer for the thorough and constructive comments. In what follows, we present detailed clarification addressing the questions raised by the reviewer. In the final version of the paper, we will also have more explanation on the assumptions.
>
> 1. What are the user strategies, $\mathbf{q}$?
>
> It is shown that players with longer memories do not have any advantage over the ones with shorter memory when the same game setup is repeated (Press and Dyson, 2012). Thus, without any loss of generality, we assume that the user strategy is memory-1: The probability of user $k$ taking the cooperative action ($A_{k,t} = c$) in iteration $t$ is conditioned on the joint action pair of the previous iteration, $(A_{k,t-1},B_{k,t-1})$. In particular, $q_{1}, q_{2}, q_{3}$ and $q_{4}$ denote the probabilities of cooperation as follows:
> \begin{align*}
> 	\mathbb{P} (B_{k,t} = c | A_{k,t-1} = C, B_{k,t-1} = c) & = q_1, \\\\
> 	\mathbb{P} (B_{k,t} = c | A_{k,t-1} = C, B_{k,t-1} = d) & = q_2, \\\\
> 	\mathbb{P} (B_{k,t} = c | A_{k,t-1} = D, B_{k,t-1} = c) & = q_3, \\\\
> 	\mathbb{P} (B_{k,t} = c | A_{k,t-1} = D, B_{k,t-1} = d) & = q_4.
> \end{align*}
> The user's strategy vector is defined as $\mathbf{q} = [q_1,\ q_2,\ q_3,\ q_4]$. Our results indicate that different user strategy $\mathbf{q}$ choices can impact the convergence rate of the algorithm.
>
> 2. Are Assumptions 1 and 2 necessary conditions?
>
> Assumption 1 and 2 are introduced only for the analysis of the proposed stochastic gradient descent with strategic users (SGD-SU) algorithm. They are only sufficient conditions.
>
> 3. Why is Assumption 2 introduced?
>
> The strategies of the user and the FC, $\mathbf{q}$ and $\mathbf{p}$, imply a Markov state transition matrix, $\Omega$ (Eq.3). Let $\Lambda_1$ denote the initial state distributions of the repeated games. If the FC was able to employ $\mathbf{p}$ without any errors, the state distributions of the games at iteration $t$ would be found as $\Lambda_t = \Lambda_1 \Omega^{t-1}$. Nevertheless, the FC cannot directly observe the user actions and cannot directly employ $\mathbf{p}$. As explained in Section 3.3 and Lemma 1, this incurs an additive perturbation on the state distributions:
> \begin{equation*}
> \tilde{\Lambda}_{t} = \Lambda_t + \Lambda_1 \sum^{t-1}\_{i=1} V_i \Omega^{i-1} \Omega^\bot \Omega^{t-1-i}.
> \end{equation*}
>
> As it is discussed in Section 3.4, the gradient estimator $\hat{m}_t$ can be decomposed as follows:
> \begin{equation}
> 	\hat{m}_t (\cdot) = m_t(1+\zeta_t) + \mathcal{E}_t, \quad (14)
> \end{equation}
> where $\mathcal{E}_t$ is the noise term and $\zeta_t$ is the bias term
> \begin{equation}
> 	\zeta_t = \frac{1}{m_t} \Big( \mathbb{E}_t[\hat{m}_t] - m_t \Big) = \frac{\tilde{\Lambda}_t \mathbf{q}^\top}{\Lambda_t \mathbf{q}^\top} -1. \quad (15)
> \end{equation}
> If the gap between $\tilde{\Lambda}_t$ and $\Lambda_t$ is very small; then, the estimation bias term becomes insignificant. Otherwise, the estimation bias may significantly hamper the convergence of the algorithm.
>
> Assumption 2 implies the following bound on the estimation bias term:
> \begin{equation}
> 	|\zeta_t| < H_T  < 1, \ \forall t \in \{1,\dots, T\}.
> \end{equation}
> Capitalizing on this bound, we are able to derive the convergence result in Theorem 1 in terms of $H_T$. Theorem 2 gives further details about $|\zeta_t|$.
>
> 4. How restrictive is Assumption 2
>
> In Assumption 2, the first condition, $\Lambda_t \mathbf{q}^\top \geq 0.5$ for every $t \in \{1, \dots, T\}$, is very mild in the sense that it only requires the probability of user cooperation dictated by $\mathbf{p}$ and $\mathbf{q}$, in the absence of perturbations, is larger than 0.5. In the Section 4 Experiments section, the convergence performance of the proposed algorithm is evaluated when the users employ different strategies. When the users employ the Tit-for-tat strategy, this condition is violated and the algorithm still performs well  Further details on this may be found in Section 4 Experiments and Appendix I Experiments. Please also see our response to Reviewer VAjM (Question 5).
>
> By definition, we have $1 \geq \tilde{\Lambda}\_t \mathbf{q}^\top \geq 0$. Thus, if $\Lambda_t \mathbf{q}^\top \geq 0.5$; then there exists a constant $H_T$ such that
> \begin{equation*}
> |\zeta\_t| = \bigg | \frac{\tilde{\Lambda}\_t \mathbf{q}^\top}{\Lambda_t \mathbf{q}^\top} - 1 \bigg | < H_T \leq 1, \forall t \in \{1, ..., T\}.
> \end{equation*}
> Under this assumption, we derive the convergence rate in Thm 1:
> \begin{equation}
> 	\mathbb{E} \left[ \frac{1}{T} \sum_{t=1}^T \left \| \nabla F (\theta_t) \right\|^2 \right] \leq \frac{L E_T}{K (1-H_T)} + \frac{2 ( F(\theta_1) - F_{\mathrm{inf}} )}{\bar{\eta} T (1-H_T)}. \quad (19)
> \end{equation}
> When $H_T$ is close to 0, the algorithm performs similar to the basic minibatch SGD. However, the limit of the right hand side of the inequality (19) equals $\infty$ as $H_T$ approaches 1. Thus, the right hand side of the inequality (19) is not well-defined when $H_T = 1$. To fix this issue, we introduce the second condition,  $\tilde{\Lambda}_t \mathbf{q}^\top > 0$ which states the probability of user cooperation is always greater than 0. Under Assumption 2, it follows that
> \begin{equation*}
> |\zeta_t| < H_T < 1.
> \end{equation*}
> In practice, the second condition of Assumption 2 is not essential. When $H_T$ is close to 1, the bound on the convergence rate becomes very loose. To this end, we introduce Theorem 2 which characterizes the gradient estimation bias term $\zeta_t$ in further detail. Please also see our response to Reviewer AUVR for detailed discussion of Theorem 2.
>
> 5. $\mathcal{Q}$-Function
>
> $\mathcal{Q}$-function is the tail distribution function of the standard Gaussian distribution.
> Equivalently, $\mathcal{Q}(x)$ can be defined as the probability that a standard normal variable $X \sim \mathcal{N}(0,1)$ takes a value larger than $x$:
> \begin{equation}
> \mathcal{Q} (x) = \mathbb{P} (X > x) =  \frac{1}{\sqrt{2\pi}} \int_{x}^\infty \exp^{- u^2/2} \mathrm{d} u.
> \end{equation}
> In the final version of the paper, we will include the definition of Gaussian-$\mathcal{Q}$ function.
>
> 6. Why is Assumption 1 introduced?
>
> As discussed in Section 3.2, if the fusion center (FC) directly employs the ZD strategies without any adaptation, the classification errors yield an adverse disturbance on the dynamics of the repeated games (Remark 4 and Eq. 10). The adaptive strategies are proposed to cancel out this adverse disturbance. The challenge for the FC is that this adaptation step (Eq. 9) requires the knowledge of the detection probability, $\Phi_t$. As it can be seen from Eq. 8, $\Phi_t$ is a function of $m_t$ (which is unknown to the FC) and cannot be exactly calculated. Instead, the FC forms her estimate of the detection probability using $\hat{m}_t$ and employs it in the adaptation step. This yields an additive perturbation on the state transition matrix:
> \begin{equation}
> \tilde{\Omega}_t = \Omega + V_t \Omega^\bot \text{ with }
> V_t := \frac{\hat{\Phi}_t - \Phi_t}{\hat{\Phi}_t -\Psi_t} \text{ and } \Omega^\bot := (p_1 - p_2) \mathbf{q}^\top [-1 \ \ 0\ \ 1\ \ 0]. \quad (12)
> \end{equation}
>
> In comparison to Eq. 10, the additive perturbation term in Eq. 12 is less severe: The adverse effects of the misclassification of the user actions is mitigated if the estimated detection probability is "sufficiently" accurate.
>
> Nevertheless, these additive perturbations directly appear in the state transition rules. In general, an exact characterization of the state distributions is intractable:
> \begin{equation*}
> \tilde{\Lambda}\_{t + 1}  =  \tilde{\Lambda}\_{t} ( \Omega  +  V_t \Omega^\bot)  =  \Lambda_1 ( \Omega  + V_1 \Omega^\bot) ( \Omega + V_2 \Omega^\bot) \dots ( \Omega + V_t \Omega^\bot)
> \end{equation*}
> Under Assumption 1, in Lemma 1, the scalar term $V_t$ can be bounded and the following result is obtained:
> \begin{equation*}
> \tilde{\Lambda}_t = \Lambda_t + \Lambda_1 \sum\_{i=1}^{t-1} V_i \Omega^{i-1} \Omega^\bot \Omega^{t-1-i}. \quad (13)
> \end{equation*}
> Using Lemma 1, we are able to derive a high probability bound on $\zeta_t$ in Theorem 2. Note that Theorem 1 does not depend on Lemma 1 and Assumption 1.
>
> 7. Typos in Approximation 1.
>
> We apologize for the typos in the statement of Assumption 1. The correct statement should be as follows:
> \begin{equation*}
> \|\hat{m}_t\| \geq \mathrm{max} \Big \\{ 2 \sqrt{\mathrm{Ray}(\Sigma_t, \hat{m}_t)}, 2 \sqrt{\mathrm{Ray}(\Xi_t, \hat{m}_t)}, \sqrt{ \big | \hat{m}_t^\top (m_t - \hat{m}_t ) \big |}  \Big \\}
> \end{equation*}
> As the reviewer indicated, $\sqrt{ \hat{m}_t (m_t - \hat{m}_t)}$ is not well-defined. This error was later corrected in the Supplementary material version of our paper which also contains the Appendices.
>
> 8. Further details on Assumption 1:
>
> In Section 3.3, $V_t$ is defined in Eq.12, It can be computed as follows:
> \begin{equation*}
> V_t = \frac{\hat{\Phi}_t - \Phi_t}{\hat{\Phi}_t - \Psi_t}
> \end{equation*}
> where $\Phi_t$ is the detection probability of the classifier
> \begin{equation*}
> \Phi_t = 1 - \mathcal{Q}\left( \frac{ \frac{\hat{m}_t(m_t - \hat{m}_t)}{\|m_t\|} + \frac{1}{2} \| \hat{m}_t \|}{\sqrt{\mathrm{Ray} (\Sigma_t, \hat{m}_t)}} \right),
> \end{equation*}
> $\Psi_t$ is the false alarm probability of the classifier
> \begin{equation*}
> \Psi_t = \mathcal{Q} \left( \frac{ \frac{1}{2} \| \hat{m}_t\| } { \sqrt{ \mathrm{Ray} (\Xi_t, \hat{m}_t) } } \right)
> \end{equation*}
> and $\hat{\Phi}_t$ is the estimated detection probability
> \begin{equation*}
> \hat{\Phi}_t = 1 - \mathcal{Q}\left( \frac{ \frac{1}{2} \| \hat{m}_t \|}{\sqrt{\mathrm{Ray} (\Sigma_t, \hat{m}_t)}} \right) .
> \end{equation*}
> Under Assumption 1, we have that $| \hat{\Phi}_t - \Phi_t | \leq 0.15$ and $| \hat{\Phi}_t - \Psi_t | \geq 0.68$. Thus, ignoring higher order $V_t$ terms is a sensible approximation. Without this step, the convergence analysis of the algorithm becomes intractable and Theorem 2 cannot be derived.

---

### Official Review · Reviewer_UwUw · 2021-07-14

**Rating:** 6
**Confidence:** 3

**Summary:**

The paper studies a federated learning scheme for training machine learning models using gradients provided by a set of strategic users. At each iteration, each user gets a mini-batch of samples, computes her gradient, and decides whether to report it to the fusion center (FC) or to report a random signal.  The authors propose a strategy for the FC which incentivizes users to truthfully report their gradients and ensure convergence guarantees. Results are validated experimentally on publicly available datasets.

**Limitations And Societal Impact:**

I don't see potential negative societal impact for this work.

**Main Review:**

Federated learning is a relevant problem in machine learning and I believe this work positively contributes to the existing literature.
The work adds both novel theoretical contributions and experimental results. Moreover, it is clearly written. Hence, my acceptance score.  However, I have a few comments for the authors:

- It would be nice to have related works discussed in the main paper, clearly highlighting the differences in terms of set-ups and assumptions considered. I believe the problem of federated learning can be tackled from a lot of different perspectives and it would be nice to have this clear from the first sections. A lot of technical definitions and details could be moved to the Appendix instead (e.g., Equation 20).

- In the experiments, only a static strategy vector is considered for the users. However, it would be natural to consider users who adapt their strategy online. Is there a particular reason why this was not considered? This would test whether the FC algorithm actually steers the agents to collaborate in the long run.

**Time Spent Reviewing:**

3

---

> ### Author Response · Authors · 2021-09-05
> **Author Response to Reviewer UwUw**
>
> The authors would like to express sincere thanks to the reviewer for their comments. We would like to apologize for the length of time it has taken us to reply due to our own extenuating circumstances. Nonetheless, we value the feedback provided and find it extremely important to respond to, given the points raised and the time the reviewer has spent on these comments. Thank you very much for your understanding.
>
> 1. We thank the reviewer for the suggestion regarding the Related Works section. We believe this is a very sensible suggestion. In the current version of the paper, the Related Works section is relegated to Appendix. In the final version of the paper, we can move this section to the main paper while moving some of the technical details such as Eq. (20). Revised Related Work section can be found as follows.
>
> Related Work
>
> Game-Theoretical Approaches in Machine Learning
>
> There are several papers that study game theoretical approaches for statistical inference and estimation in the presence of strategic agents [4,5,7,8,11,12,20,24]. There are three key differences between our work and these studies: (i) the nature of the collected signals, (ii) repeated interactions between the center and the users, and (iii) the role of payments.
>
> (i) In these studies, strategic users are the data sources. The goal can be the inference of a model [4,5,8,11,12] or the estimation of a parameter of interest [7,20] from the agents' data. Alternatively, in [24], the authors propose an interesting system where the users' data is the machine learning models. While the users in our study are also strategic data sources, they never directly or indirectly reveal their raw data. The fusion center only collects the stochastic gradients from the users to train a machine learning model.
>
> (ii) All these works consider single-stage games where the center and the users interact only once. In our study, the fusion center and the users interact repeatedly. The behaviors of the users intertwines with the stochastic gradient updates. Thus, one of our primary goals is to evaluate the impact of the repeated games on the convergence performance of stochastic gradient descent algorithm in the presence of strategic users.
>
> (iii) Several papers consider scenarios where users as data sources incur a privacy cost by releasing their data [7,11]. Relevant to our work, in these papers, the center uses rewards in the form of monetary payments to encourage users to reveal their private data. These studies consider the mechanism design problem and focus on the trade-offs between payment and accuracy or payment and privacy. One of our primary objectives is the design of a repeated game strategy, for the fusion center, based on the zero-determinant strategies. In contrast to the mechanism design approach, in our study, the fusion center is allowed to reciprocate against non-cooperative users based on the state of the repeated game. In [5,8,12], the payoffs of the users depend on the outcome of the estimation process. This line of research focuses on ``strategy-proof" algorithms which are robust against manipulated inputs, without using monetary payments.
>
> A recent related work [34] explores a federated learning setting with independent and self-interested participants. The center collects the model updates from the users, evaluates the quality of the reported model updates and rewards them accordingly. In contrast to our work, this study also focuses on the economics of a federated learning system at a single iteration rather than the impact of untruthful reporting on the overall performance of the learning scheme throughout the entire training process. In [30], a multi-player game is proposed to study the reactions of strategic participants, in various federated learning ecosystems, for various incentive mechanisms. However, the scope of this study is limited to the development of an interactive user interface to collect data for future experimental studies.
>
> Repeated Games
>
> The pioneering work of Press and Dyson [33] shows that it is possible for a player to unilaterally impose a linear relationship between their and the opponent’s payoff employing “zero-determinant” (ZD) strategies in a 2x2 repeated game. In this study, both players can observe the action of their opponent in a perfect environment without any noise. In [15], the ZD strategies in noisy games is examined under the assumption that the players know the time-invariant error distribution. In our paper, however, the fusion center cannot directly receive any
> (noisy or noiseless) observation of the user action. In order to address this key difficulty, we devise a user action classifier for the fusion center based on the collected reported gradients. Further, we propose an adaptive zero-determinant (ZD) strategy conditioned on the outcomes of the user action classifier. Due to the nature of the data driven gradient updates, the user action classification incurs time-varying stochastic errors, which adds another non-trivial complexity. Thus, our work can be viewed as a contribution to the literature of repeated games, as well as a contribution to distributed learning.
>
> Byzantine-Resilient Machine Learning
>
> In the presence of malicious devices, the robustness issues in distributed learning has received much attention. In these studies, it is assumed that good devices dominate the entire set of devices and it is proposed that fault-tolerant algorithms can trim the outliers from the candidates [1, 2, 9, 35]. The basic goal of these studies differs from ours, since we consider a game-theoretic setting in which the users are utility-driven who have the ability to formulate strategies to choose their actions, cooperative or defective, which can depend on the outcome of previous interactions with the FC.
>
> 2. In this study, one of our goals is to explore the impact of different user strategy choices on the convergence performance of the stochastic gradient descent algorithm with strategic users. To this end, our analytical results are summarized in Theorems 1 and 2. It is nontrivial to generalize these findings to the scenarios in which the users are evolutionary players who adjust their strategy according to some optimization scheme. Nevertheless, we thank the reviewer for this suggestion. In the final version of the paper, we will include experimental results to test whether there exist evolutionary paths for the users that lead to cooperation in the long run.
>
> We believe this is a very interesting research direction to explore the performance of the proposed algorithm in the presence of evolutionary players and we will work on this in our future works.

---

### Official Review · Reviewer_VAjM · 2021-07-16

**Rating:** 6
**Confidence:** 3

**Summary:**

The paper formulates the interaction between the fusion center (FC) and users in distributed learning (or federated learning) from the perspective of game theory. In order to encourage the users to provide effective information to the FC for model update, the FC adopt zero-determinant (ZD) strategy, associated with which there are some practical issues addressed by the authors. Theoretical analysis on the convergence of the proposed algorithm is presented.

**Limitations And Societal Impact:**

No negative societal impact.

**Main Review:**

The paper formulates the interaction between the fusion center (FC) and users in distributed learning (or federated learning) from the perspective of game theory. In order to encourage the users to provide effective information to the FC for model update, the FC adopt zero-determinant (ZD) strategy, associated with which there are some practical issues addressed by the authors. Theoretical analysis on the convergence of the proposed algorithm is presented. The formulation and the results are interesting; however, there are some concerns from the reviewer:

1.	What’s the motivation of the users submitting random noninformative signal to the FC instead of just submitting nothing when they decide not to cooperate. Submitting random noninformative signal does not benefit the users but harms the FC.
2.	Please explain the state transition matrix $\Omega$ in eq. (3), why it is constructed as it is and why the states are supposed to evolve according to it. Just below eq. (3), the authors use \bar{\Omega} to denote the transition matrix, what is the difference between $\Omega$ and $\bar{\Omega}$, or it is just a typo?
3.	The authors are expected to clearly state the goal of users in playing the game, namely, in each iteration, the users maximize what, the expected long-term reward, or the reward of this iteration. Based on the objective of the users, the author shall show that the users’ best response is q* = [1, 1, 1, 1]^T given that the FC adopts ZD strategy.
4.	In the Experiments section, the figure referred is Fig 2 rather than Fig 4. SGSU is SGD-SU, typo? Does Pavlov mean the win-stay-lose-switch? If it does, please just use one of these two phrases throughout the paper.
5.	From Figure 2(a) and (b), it is shown that the optimality gap is the smallest when the users take tit-for-tat strategy. Does this mean the FC prefers tit-for-tat instead of full-cooperation in order to obtain a favorable solution to optimization problem (1)? This is not consistent with the game, where the FC prefers full-cooperation?
6.	Since the problem is formulated as a repeated game, the experiments shall show whether the strategy of the users converges exactly to q* starting from any initial state and action.
7.	The reviewer believes (24) admits a closed-form solution, and the computational complexity of (24) is the same as that of (6). Please check this.


**Time Spent Reviewing:**

8

---

> ### Author Response · Authors · 2021-08-11
> **Author Response to the Reviewer VAjM**
>
>
> The authors would like to express sincere thanks to the reviewer for the thorough and constructive
> comments. In what follows, we present detailed clarification addressing the questions raised by the reviewer. We first start with the second question since some of the notation introduced in our response is also helpful to understand the first question.
>
> 2. We apologize for the typo. In Line 91 (right after Eq.3), it should be written as follows: Let $\Lambda^*$ be the stationary vector of the transition matrix $\Omega$, i.e., $\Lambda^* = \Lambda^* \Omega$.
>
> The probability of user $k$ taking the cooperative action ($A_{k,t} = c$) in iteration $t$ is conditioned on the joint action pair of the previous iteration, $(A_{k,t-1},B_{k,t-1})$. In particular, $q_{1}, q_{2}, q_{3}$ and $q_{4}$ denote the probabilities of cooperation:
> \begin{align*}
> \mathbb{P} (B_{k,t} = c | A_{k,t-1} = C, B_{k,t-1} = c) & = q_1, \\\\
> \mathbb{P} (B_{k,t} = c | A_{k,t-1} = C, B_{k,t-1} = d) & = q_2, \\\\
> \mathbb{P} (B_{k,t} = c | A_{k,t-1} = D, B_{k,t-1} = c) & = q_3, \\\\
> \mathbb{P} (B_{k,t} = c | A_{k,t-1} = D, B_{k,t-1} = d) & = q_4.
> \end{align*}
> The user's strategy vector is defined as $\mathbf{q} = [q_1,\ q_2,\ q_3,\ q_4]$. The strategy vector of the FC is defined as $\mathbf{p} = [p_1,\ p_2,\ p_3,\ p_4]$ where $p_{1}, p_{2}, p_{3}$ and $p_{4}$ denote the conditional probabilities of cooperation:
> \begin{align*}
> 	\mathbb{P} (A_{k,t} = C | A_{k,t} = C, B_{k,t-1} = c) & = p_1, \\\\
> 	\mathbb{P} (A_{k,t} = C | A_{k,t} = C, B_{k,t-1} = d) & = p_2, \\\\
> 	\mathbb{P} (A_{k,t} = C | A_{k,t} = D, B_{k,t-1} = c) & = p_3, \\\\
> 	\mathbb{P} (A_{k,t} = C | A_{k,t} = D, B_{k,t-1} = c) & = p_4.
> \end{align*}
> Note that the action of the FC is conditioned on her action in $t-1$ and the user's action in $t$. This is due to the sequential nature of the game: The FC takes her actions after collecting the reported gradients of the users.
>
> We define the joint action pair $(A_{k,t}, B_{k,t})$ as the state of the repeated game. The elements of the state transition matrix $\Omega$ are the transition probabilities from different states. To illustrate, we can calculate the probability of moving from state $(D,c)$ to $(C,d)$. Observe that
> \begin{align*}
> \mathbb{P} & (\ (A_{k,t+1}, B_{k,t+1}) = (C, d) | (A_{k,t}, B_{k,t}) = (D, c)\ ) \\\\
> & = \mathbb{P} (A_{k,t+1} = C | B_{k,t+1} = d, (A_{k,t}, B_{k,t}) = (D, c)\ ) \ \mathbb{P} (B_{k,t+1} = d | (A_{k,t},  B_{k,t}) = (D, c) \ ), \\\\
> & = \mathbb{P} (A_{k,t+1} = C | B_{k,t+1} = d, A_{k,t} = D )  \mathbb{P} (B_{k,t+1} = d | (A_{k,t},  B_{k,t}) = (D, c)\ ), \\\\
> & = p_4 (1-q_3) = \Omega_{3,2}.
> \end{align*}
> The first equality follows from the chain rule of probability, and the rest follows from the definition of the strategy vectors, $\mathbf{p}$ and $\mathbf{q}$. In a similar fashion, the other elements of $\Omega$ can also be computed.
>
> 1. As discussed in Section 2, the original zero-determinant (ZD) strategy (Press and Dyson, 2012) is a powerful tool to incentivize user cooperation. By employing this class of strategies, the FC can enforce a linear relation between the expected payoffs. It can be in the form of setting the expected payoff of the user or demanding an extortionate share of payoffs. If the defective action of the users is not sending any signal, the fusion center (FC) can directly observe whether user $k$ is cooperative or defective. In this case, she can essentially employ any strategy from the class of ZD strategies.
>
> When the users send uninformative signals, the FC cannot directly observe the user actions and cannot directly employ the ZD strategy. To this end, we propose a novel approach where the FC first classifies the user actions (with possible errors) and then adapt her strategy according to the accuracy rate of the user action classification. We should remark that, under this approach, the feasible set of ZD strategies is smaller than the original class of ZD strategies. In particular, the aforementioned adaptation step can be applied if the following condition holds (Section 3.2, Eq. 9):
> \begin{equation*}
> 	\frac{\Phi_t}{\Psi_t} \geq \frac{p_1}{p_2} \text{ and } \frac{\Phi_t}{\Psi_t} \geq \frac{p_3}{p_4},
> \end{equation*}
> where $\Phi_t$ and $\Psi_t$ are the detection and false alarm probabilities associated to the user action classification.
>
> To understand how this restricts the FC, consider an example where the FC chooses her ZD strategy as: $\mathbf{p} = [1,\ \epsilon,\ 1,\ \epsilon]$. In this strategy, she always rewards a user when the user is cooperative. With probability $\epsilon$, she may still reward the user even if the user is defective. The adaptation step is only feasible if $\epsilon \in [\Psi_t/\Phi_t, 1]$ and the inevitable classification errors forces the FC to choose more ``forgiving" ZD strategies. Please also see our response to the 3rd and 5th questions.
>
> 3. That is a very good question. In 2x2 games, it is shown that players with longer memories do not have any advantage over the ones with shorter memory when the same game setup is repeated (Press and Dyson, 2012). Thus, without any loss of generality, we assume the user employs a memory-1 strategy, denoted by $\mathbf{q}$.
>
> In the hypothetical scenario where the FC can directly observe the user actions, she can become the dominant player. In this case, the best strategy for the user which maximizes his expected payoff in the steady-state is full cooperation. If a user challenges the dominance of the FC, she may discard that user, given that there are sufficiently many users in the system.
>
> Unlike this hypothetical scenario, the FC cannot directly observe the user actions. For this case, the characterization of the best user strategy is a very challenging problem and it is still unresolved in this study. Instead, we primarily focus on the impact of strategic user actions on the convergence performance. In Theorem 1 and 2, the analysis is provided for generic $\mathbf{q}$.
>
> It should be noted that certain user strategies are observed to be more resilient against the ZD strategies of the FC because the proposed user action classification scheme solely relies on the reported gradients of the users. In the experiments, we observe that the Pavlov users exploit this weakness. That is a unique phenomenon due to the nature of the distributed learning problem. We are not aware of any other study on ZD strategies with a similar observation. Please also see our response to the 5th question.
>
> 4. We apologize for the typo and thank the reviewer for the corrections. In Section 4, the figure should be referred as Fig 2 and the algorithm should be referred as SGD-SU. Win-stay-lose-switch is another name for the Pavlov strategy. In the final version of the paper, we will only use "Pavlov strategy".
>
> 5.  We thank the reviewer for the careful observation. We noticed that there is a minor error in the Legend of Figure 3b in Appendix I - Experiments. The labels for Full Coop and Tit-for-Tat should be swapped. The same plot is already included in the main body of the paper and its legend is accurate. Nevertheless, from Fig 2a and 3a, it can be seen that the performance of the SGD-SU is very similar when the user employs a Full Coop, Tit-for-Tat, Stubborn or Coin Toss strategy. This is an unexpected observation since ratio of the cooperative users are very different in these scenarios. To understand this, we need to get into the details of the convergence analysis.
>
> Recall that the gradient estimator is introduced in Eq. 6:
> \begin{equation*}
> \hat{m}_t (\mathbf{Y}_t) = \frac{1}{ \Lambda_t \mathbf{q}^\top}  \frac{1}{K} \sum Y_k
> \end{equation*}
> The multiplier, $1/(\Lambda_t\mathbf{q}^\top)$, is introduced as a correction term to cancel the bias due to the defective users. Without any perturbations, the probability of user $k$ taking the cooperative action can be found as $\Lambda_t \mathbf{q}^\top$ and the proposed estimator is unbiased. In the presence of perturbations, the true probability of user $k$ taking the cooperative action is not exactly equal to $\Lambda_t \mathbf{q}^\top$. This causes a bias (Eq. 14):
> \begin{align*}
> \hat{m}_t  = m_t(1+\zeta_t) + \mathcal{E}_t, \text{ where } \zeta_t = \frac{\tilde{\Lambda}_t \mathbf{q}^\top }{\Lambda_t \mathbf{q}^\top} -1.
> \end{align*}
>
> When the users employ the Tit-for-Tat, Stubborn or Coin Toss strategy, they cooperate less often compared to the Full Coop strategy. In these cases, the variance of the noise term $\mathcal{E}_t$ increases and the gradient estimator becomes more noisy. That being said, as long as the true probability of user cooperation is close to the theoretical value $\Lambda_t \mathbf{q}^\top$, the bias term will be very small.  From Fig2(c) and Fig2(d), we see that true probability of user cooperation is very similar to the theoretical values. As a result, the bias term is very small, $\zeta_t \approx 0$.
>
> For the Pavlov users, we observe a very different behavior. From Fig2(c-d), we see that true probability of user cooperation is significantly smaller than the theoretical value, 0.67. Thus, the gradient estimator is biased and this significantly hampers the performance of the algorithm. Consequently, the convergence of the algorithm can be affected by both the bias and the noise in the gradient estimation. In this set of experiments, the impact of the bias is more prominent. Therefore, the Pavlov strategies are not preferable. Further discussion is also provided in Appendix I - Experiments.
>
> 6. We thank the reviewer for this suggestion. In the final version of the paper, we will include experimental results with different initial state distributions.
>
> 7. To the best of our knowledge, in general, there is no explicit formula of the geometric median (24) unlike the centroid or center of mass. The geometric median can be approximately computed in an efficient manner using the Weiszfeld’s algorithm (refs [36], [37]).

---

### Official Review · Reviewer_AUVR · 2021-07-25

**Rating:** 5
**Confidence:** 3

**Summary:**

This paper proposed an incentive mechanism for a distributed learning setting. The users are encouraged but not obliged to provide the gradient. The results on convergence and an upper bound on the estimation bias of the gradient were shown.

**Main Review:**

I am mostly concerned about the setup (line 64-line 74).

“In iteration t, when a user cooperates, he provides an information gain R to the FC at his privacy cost VUR with 0”. It is unclear to me how the authors quantifies the privacy cost and why it is related to the information gain R via a factor V_U between 0 and 1. It seems to me that the information gain R was not used in Algorithm 1. Would the authors explain how R is used? Moreover, the information gain R should vary every time. Is it the case in the paper’s setup?

“The FC is cooperative (Ak,t=C) if she makes a payment r to the user at her cost rVFC with 0”. Why is the cost of the FC rV_{FC} related to V_{FC} via a factor V_FC between 0 and 1? Would the authors explain the intuition behind this?

line 72-73: the authors did not specify how each entry in a payoff vector is ordered.

It would be beneficial to discuss when Phi_t/Psi_t>=p1/p2, Phi_t/Psi_t>=p3/p4 are satisfied and if they are always satisfied (line 151).

I like it that Theorem 2 gives a bound on \zeta_t. But there are still some unclear things.

how large is $\sum_{i=1}^{t-1} \alpha_i^2$ compared to $\eta^2$? The authors should establish that the bound (22a) would not be vacuous. Recall in line 61 that the norm of the gradient is <= \ell. This bound is vacuous if the upper bound is just close to \ell. The authors may want to compute the expected value of \zeta_t and then check if the bound is vacuous.

What is $\Delta_{i,t}$?

The current form looks pretty complicated. The authors may want to do some asymptotic analysis and present the asymptotic order of each term so that the presentation of the results can also be simplified.

**Time Spent Reviewing:**

3

---

### Decision · Program_Chairs · 2021-09-27

**Decision:**

Reject

**Comment:**

Summary:

The paper considers a setting motivated by federated learning, where a fusion center (FC) seeks to minimize some loss function over k users, but has to rely on stochastic gradients reported by the users. This is set up as a repeated game in which users can either cooperate (and send the true stochastic gradient) or defect and send a noisy signal.

The main result is a stochastic gradient descent-strategic users (SGD-SU) algorithm, which builds on ideas of the zero-determinant strategy for repeated 2x2 games (Press and Dyson, PNAS, 2012).
The paper presents theoretical results on the effectivity of their algorithm (Theorem 1 and Theorem 2), as well as experimental results (on the KDD Cup 04 dataset and the MNIST dataset).

Recommendation:

All reviewers were a bit lukewarm on this paper. They generally liked the basic question/problem, but also expressed a variety of concerns regarding the concrete modelling assumptions.
My evaluation is closest to that of reviewer mQAc, which felt that the assumptions (Assumption 1 and Assumption 2) are a bit artificial, and that it's a bit unclear what information is gained from the theoretical bounds.
Overall, I don't see enough enthusiasm here to recommend the paper for acceptance.